# Modulation of the 5-Lipoxygenase Pathway by Chalcogen-Containing Inhibitors of Leukotriene A_4_ Hydrolase

**DOI:** 10.3390/ijms24087539

**Published:** 2023-04-19

**Authors:** Tarvi Teder, Stefanie König, Rajkumar Singh, Bengt Samuelsson, Oliver Werz, Ulrike Garscha, Jesper Z. Haeggström

**Affiliations:** 1Division of Physiological Chemistry II, Department of Medical Biochemistry and Biophysics, Karolinska Institutet, 17177 Stockholm, Sweden; 2Department of Pharmaceutical/Medicinal Chemistry, Institute of Pharmacy, Greifswald University, 17489 Greifswald, Germany; 3Department of Pharmaceutical/Medicinal Chemistry, Institute of Pharmacy, Friedrich Schiller University Jena, 7743 Jena, Germany

**Keywords:** 5-lipoxygenase, aminopeptidase, inflammation, leukotriene A_4_ hydrolase

## Abstract

The 5-lipoxygenase (5-LOX) pathway gives rise to bioactive inflammatory lipid mediators, such as leukotrienes (LTs). 5-LOX carries out the oxygenation of arachidonic acid to the 5-hydroperoxy derivative and then to the leukotriene A_4_ epoxide which is converted to a chemotactic leukotriene B_4_ (LTB_4_) by leukotriene A_4_ hydrolase (LTA_4_H). In addition, LTA_4_H possesses aminopeptidase activity to cleave the N-terminal proline of a pro-inflammatory tripeptide, prolyl-glycyl-proline (PGP). Based on the structural characteristics of LTA_4_H, it is possible to selectively inhibit the epoxide hydrolase activity while sparing the inactivating, peptidolytic, cleavage of PGP. In the current study, chalcogen-containing compounds, 4-(4-benzylphenyl) thiazol-2-amine (ARM1) and its selenazole (TTSe) and oxazole (TTO) derivatives were characterized regarding their inhibitory and binding properties. All three compounds selectively inhibit the epoxide hydrolase activity of LTA_4_H at low micromolar concentrations, while sparing the aminopeptidase activity. These inhibitors also block the 5-LOX activity in leukocytes and have distinct inhibition constants with recombinant 5-LOX. Furthermore, high-resolution structures of LTA_4_H with inhibitors were determined and potential binding sites to 5-LOX were proposed. In conclusion, we present chalcogen-containing inhibitors which differentially target essential steps in the biosynthetic route for LTB_4_ and can potentially be used as modulators of inflammatory response by the 5-LOX pathway.

## 1. Introduction

Leukotrienes (LTs) are a group of potent bioactive lipid mediators synthesized from arachidonic acid (AA) through consecutive enzymatic steps by 5-lipoxygenase (5-LOX) and leukotriene A_4_ hydrolase (LTA_4_H) or leukotriene C_4_ synthase (LTC_4_S) [1,2]. LTs are produced in human myeloid cells and act in an autocrine or paracrine manner, exerting their functions through the corresponding receptors on target cells [3].

5-LOX (EC 1.13. 11.34) is a non-heme iron-containing dioxygenase that catalyzes stereo- and regio-selective insertion of molecular oxygen into AA which results in the formation of 5*S*-hydro(peroxy)eicosatetraenoic acid (5*S*-H(p)ETE) and the unstable leukotriene A_4_ epoxide, LTA_4_ [4] (Figure 1). The ratio between 5*S*-HETE and LTA_4_ depends on several intracellular modulating factors, such as the substrate concentration and the interaction between 5-LOX and Five-Lipoxygenase-Activating Protein (FLAP) or Coactosin-Like Protein (CLP). In human leukocytes, the 5-LOX pathway is activated upon certain physiological and pathological stimuli which elevate the levels of intracellular Ca^2+^ and increase phosphorylation events by mitogen-activated protein kinases [4]. The 5-LOX structure of the stable variant alone [5] and in a complex with two inhibitors, AKBA and NDGA, has been elucidated [6]. Although the 5-LOX pathway contributes to the formation of pro-inflammatory LTs, the interplay between 5-LOX and 15-LOX results in the formation of trihydroxy lipid mediators, lipoxins, that are involved in the resolution of inflammation [7,8] (Figure 1).

LTA_4_H (EC 3.3.2.6) is a bifunctional zinc-containing enzyme that catalyzes the conversion of the labile LTA_4_ epoxide to the pro-inflammatory and immune-modulating lipid mediator, leukotriene B_4_ (LTB_4_) [9]. In addition to the epoxide hydrolase activity, LTA_4_H also catalyzes the peptidolytic inactivation of a chemotactic tripeptide, Pro-Gly-Pro (PGP) (Figure 1) [10,11]. This neutrophil chemoattractant is produced from the extracellular matrix by metalloproteases in response to inflammation and tissue remodeling [12]. PGP is a biomarker of chronic obstructive pulmonary disease [13] and cystic fibrosis [14]. In addition, the accumulation of PGP or its acetylated derivative, AcPGP, is linked to neutrophil-mediated acute inflammation in lungs [11,14,15]. Therefore, selective inhibition of the biosynthesis of pro-inflammatory LTB_4_ with simultaneous clearance of PGP by LTA_4_H would contribute to the resolution of inflammation. Moreover, the inhibition of LTA_4_H results in unused LTA_4_ which may potentially increase conversion to cysteinyl-leukotrienes [16], or pro-resolving mediators, such as lipoxins and resolvins [17,18].

The elucidation of high-resolution crystal structures of LTA_4_H revealed the binding mode of LTA_4_ [19] and PGP [20] in the L-shaped substrate channel with overlapping active sites (Figure 1). The channel can be divided into two parts: the wider polar region containing the substrate entry site and catalytic center, and the narrower hydrophobic tunnel accommodating long fatty acid tails. LTA_4_ is bound to the hydrophobic substrate channel in a head-to-tail (“tail first”) orientation with its epoxy group interacting with catalytically essential zinc ion and the carboxylate interacting with Arg563. PGP binds near the zinc and occupies only a small part of the hydrophobic channel. This structural information gave the opportunity to design selective inhibitors which interact with residues of the narrow hydrophobic tunnel, leaving the PGP binding site unoccupied.

Initially, only artificial peptide analogues were used to characterize several small-sized diphenylether [21] or benzylphenyl [22] inhibitors, such as 4-methoxydiphenylmethane (4-MDM), on the aminopeptidase activity of LTA_4_H. However, the discovery by Snelgrove et al., 2010 that naturally occurring endogenous PGP is a substrate for LTA_4_H, stimulated further studies on selective inhibition of the two activities of LTA_4_H. In silico screening resulted in a potent selective inhibitor of LTA_4_H, 4-(4-benzylphenyl) thiazol-2-amine (ARM1), which was co-crystallized together with a PGP analogue [20]. The three-dimensional crystal structure demonstrated that ARM1 and PGP can bind to LTA_4_H simultaneously. In addition, ARM1 selectively blocked the formation of LTB_4_ without affecting the peptidolytic cleavage of PGP [20]. To date, different ARM1 derivatives have been tested [23,24]. In addition to compounds containing the benzylphenyl moiety, polyphenolic resveratrol-containing inhibitors have been synthesized and shown to target LTA_4_H in a similar manner [25].

In the current project, new inhibitors, 4-(4-benzylphenyl) selenazol-2-amine (TTSe) and 4-(4-benzylphenyl) oxazole-2-amine (TTO) were synthesized based on the lead compound, ARM1. As thiazole moiety of ARM1 binds to the end of the substrate channel of LTA_4_H via hydrogen bonds, alterations in the polar part of the inhibitor may improve its binding properties. Specifically, the sulphur atom in the thiazole ring of ARM1 was replaced with other chalcogens, namely selenium or oxygen, resulting in the selenazole and oxazole derivatives, TTSe and TTO, respectively. Different properties of chalcogens affect the physicochemical parameters of inhibitors [26] and may result in increased selectivity and inhibition. The main objective was to assess the potency and binding properties of these inhibitors using biochemical and biophysical approaches along with elucidation of their binding modes to LTA_4_H and 5-LOX. 

## 2. Results

### 2.1. Pharmacological Properties of Chalcogen-Containing Inhibitors

Two new selective inhibitors were synthesized based on the lead molecule, ARM1 (Figure 2A, Appendix A). Based on the cytotoxicity assays, ARM1, TTSe and TTO were assessed to be safe to carry out experiments in leukocytes (Appendix A). In addition, different pharmacological and biophysical properties, including bioavailability and toxicological effects, were predicted for chalcogen-containing inhibitors (Appendix A).

### 2.2. Chalcogen-Containing Inhibitors Block the Activities of 5-LOX and LTA_4_H

ARM1, TTSe and TTO inhibited the biosynthesis of 5*S*-HETE and LTB_4_ in stimulated intact polymorphonuclear neutrophils (PMNs) (Figure 2B). All inhibitors possess higher potency on the activity of LTA_4_H (LTB_4_) than 5-LOX (5*S*-HETE). However, it should be noted that cellular LTB_4_ is produced by consecutive bioactions of 5-LOX and LTA_4_H. The IC_50_ values of ARM1, TTSe and TTO on the activity of LTA_4_H were determined as 0.6, 0.8 and 0.8 µM. In contrast, the IC_50_ values on the 5-LOX activity were higher and are presented from the best to worse as follows: ARM1, 1.7 µM; TTSe, 2.3 µM; TTO, 2.3 µM (Figure 2B).

Furthermore, the inhibition constants (Ki) with recombinant human LTA_4_H (Figure 2C) and the IC_50_ values for 5-LOX (Figure 2D) were determined. The K_i_ values for ARM1, TTSe and TTO with LTA_4_H were 1.5, 4.8 and 3.8 µM, respectively. Although the trend indicated a higher potency of ARM1, the difference in the K_i_ values did not reach statistical significance. A somewhat similar potency of inhibitors on the activity of LTA_4_H is in correlation with the effect on the biosynthesis of LTB_4_ in PMNs. In addition, it should be noted that ARM1 and TTSe do not affect the activity of recombinant soluble epoxide hydrolase (sEH) activity, whereas TTO at higher concentrations may affect metabolism of fatty acid epoxides (Appendix A). In comparison with LTA_4_H, the inhibition of recombinant 5-LOX by chalcogen-containing inhibitors demonstrated distinct inhibitory properties of ARM1, TTSe and TTO (Figure 2D). Specifically, the IC_50_ values of ARM1 and TTSe were around 6 and 20 µM, respectively, while more than 30 µM of TTO was required to achieve the half-maximal inhibitory concentration. 

ARM1 has previously been demonstrated as a selective inhibitor of the epoxide hydrolase activity of LTA_4_H, while PGP cleavage remains unaffected [20]. Similarly to ARM1, TTSe and TTO did not influence the cleavage of the PGP tripeptide by recombinant LTA_4_H (Figure 2E). 

To see the effect of the exogenous substrate on the potency of each inhibitor at low and high concentrations (0.5 µM vs 10 µM), inhibition assays were carried out with PMNs and 0–20 µM AA. The levels of LTB_4_ and 5*S*-HETE in the presence or absence of inhibitors at the corresponding concentration of AA (Appendix A) were compared and presented in Figure 3. The production of LTB_4_ by intact PMNs at the lower concentrations of inhibitors, 0.5 µM, was not affected by exogenous AA (Figure 3A–C), except for ARM1 at 20 µM of AA. The inhibition of LTA_4_H was almost complete with 10 µM of ARM1 and TTSe, while the addition of exogenous AA resulted in higher production of LTB_4_ in TTO-treated PMNs (Figure 3C). This is in correlation with the results indicating lower IC_50_ and K_i_ values with TTO. In addition, the production of 5*S*-HETE by PMNs in the presence of 0.5 µM of ARM1 was increased upon the addition of AA (Figure 3D). Compared with ARM1, 0.5 µM of TTSe and 0.5 µM of TTO did not have any inhibitory effects on the 5-LOX activity (Figure 3E,F). This is also explained by the weaker IC_50_ values with TTSe and TTO using intact PMNs or recombinant 5-LOX (Figure 2B,D). In contrast, 10 µM of chalcogen-containing inhibitors reduced 5-LOX activity significantly; however, the effect of 20 µM of AA on the levels of 5*S*-HETE was observed only with ARM1 and TTO (Figure 3D,F). Overall, these results indicate that the addition of exogenous AA has little or no effect on the inhibition of LTA_4_H by ARM1, TTSe and TTO, whereas the weaker inhibition of 5-LOX may lead to increased production of 5*S*-HETE by exogenous AA.

### 2.3. The Chalcogen-Containing Inhibitors Affect the Cleavage of Synthetic Peptides by LTA_4_H

LTA_4_H cleaves amides to release N-terminal natural [27] or unnatural amino acids [28] of short peptides with high efficiency. It has been established that ARM1 increases the peptidase activity of LTA_4_H with L-Ala-pNA and L-Val-pNa [20] which was also confirmed in the current study (Figure 4A–C). In addition, other peptide conjugates were tested (Figure 4E–H).

Reaction rates with pNa derivatives and recombinant LTA_4_H from the best to lowest were determined as follows: L-Arg-pNa, L-Ala-pNa, L-Leu-pNa, L-Pro-pNa, L-Lys-pNa and L-Val-pNa (Appendix A). The peptidase activity with AMC derivatives, L-Arg and L-Pro, was identical, 31 µmol/µg/min. It should be observed that the reaction rates with pNa and AMC substrates cannot be strictly compared due to different experimental settings.

In addition, all inhibitors increased the cleavage of hydrophobic pNa conjugates and the effect from the highest to lowest are as follows: L-Val-pNa, L-Leu-pNa and L-Ala-pNa. The aminopeptidase activity with other peptide analogues, L-Arg-pNa, L-Lys-pNa, L-Pro-pNa, L-Arg-AMC and L-Pro-AMC, except L-Pro-pNa, was concentration-dependently reduced by inhibitors. Even though PGP, L-Pro-pNa and L-Pro-AMC share the N-terminal proline, only the cleavage of L-Pro-AMC was inhibited. These observations indicate that due to distinct structural motifs, pNA and AMC derivatives are bound to LTA_4_H differently (see Section 2.4).

Based on the peptidase assay with LTA_4_H and different peptide analogues, the effective activating or inhibiting concentrations, the AC_50_ or IC_50_ values, for ARM1, TTSe and TTO were calculated (Figure 4). The lowest AC_50_ values were determined with L-Ala-pNa and L-Val-pNa substrates. The peptidase activity was inhibited with L-Arg-pNa, L-Arg-AMC and L-Pro-AMC substrates. The AC_50_/IC_50_ values with L-Pro-pNa were not determined due to the apparent effect of inhibitors (Figure 4D).

### 2.4. Selective Inhibitors Increase the Peptidase Activity of LTA_4_H with Small Peptide Analogues

In silico analysis of docked peptide analogues to LTA_4_H demonstrated a correlation between the reaction rates of LTA_4_H and the location of the cleavable peptide bond. Selective inhibitors (ARM1) prevent the binding of peptide analogues to the end of the L-shaped binding pocket (Figure 5A–D) but at the same time, they may either improve or inhibit the peptidase activity (Figure 4). Smaller hydrophobic peptide analogues are cleaved with higher rates in the presence of inhibitors due to the proximity of the peptide bond to catalytic Zn^2+^ (Figure 5A). The cleavage of longer peptide analogues, such as L-Arg-pNa, is not that effective in the presence of inhibitors, apparently due to a distant cleavable amide bond (Figure 5B). However, it should be noted that L-Arg-pNa is the best substrate for LTA_4_H without any inhibitors present. Most likely, L-Arg-pNa has stronger ionic interactions with LTA_4_H and efficient binding to the substrate channel which increases catalytic efficiency.

The cleavage rate of L-Pro-pNa is not affected by inhibitors due to a similar binding mode to the catalytic center as the native PGP (Figure 5C). However, it is not clear why the peptidolysis of L-Pro-pNa and PGP is not affected by inhibitor-induced conformational changes. It may be postulated that the pyrrolidine ring of the N-terminal proline provides catalytically efficient interaction regardless of the conformation of LTA_4_H.

In contrast, the binding of AMC derivatives to the catalytic center is sterically hindered by the bulkier AMC side group which does not allow efficient peptidolysis by LTA_4_H (Figure 5B,D). Therefore, the cleavage of different AMC derivatives, L-Pro-AMC and L-Arg-AMC, is blocked by selective inhibitors in a similar manner (Figure 4G,H).

To see how the peptide analogues bind to LTA_4_H in an open conformation, the structure of LTA_4_H containing the D375N mutation (PDB ID: 5NID) [19] was used as a template for docking simulations. All pNA substrates bind to the end of hydrophobic pocket, while the peptide analogues with the AMC group stay in between the catalytic center and the end of the substrate channel (Figure 5E,F). This may also explain identical activity values determined with L-Pro-AMC and L-Arg-AMC without any inhibitors. It should be noted that the catalytically efficient binding of substrates to LTA_4_H is a complicated multistep event which involves several conformational changes and rearrangements in the catalytic center. For instance, the catalytic center of LTA_4_H contains Tyr378, Tyr383 and Glu318 which behave as “a gate” that switches from an open to the closed position upon binding of LTA_4_ [19] and PGP [20]. The LTA_4_H D375N mutant in an open conformation was also captured with rotated Tyr378 (PDB ID: 5NIA) [19] which represents an open gate and actually blocked the binding of ligands to the end of the substrate channel (Figure 5G,H). This indicates that the catalytically productive binding mode of peptide analogues is not determined solely by the tyrosine “gate” and additional conformational changes need to take place in parallel. It is shown with the LTA_4_H D375N mutant in a complex with LTA_4_ where the gate is closed and the epoxy group of LTA_4_ is located close to the catalytic Zn^2+^. Inhibitor-bound LTA_4_H (Figure 5A–C) and LTA_4_H in a closed conformation represent the same conformational state—the LTA_4_H has a narrower substrate channel due to the domain movement and at the same time, Tyr378 and Tyr383 are faced towards the catalytic Zn^2+^ which is ready to carry out the hydrolase reaction. To conclude, chalcogen-containing inhibitors induce the conformational changes and activate the tyrosine gate which increase the peptidolysis of small hydrophobic peptide analogues.

### 2.5. Selective Inhibitors Bind to LTA_4_H in a Similar Manner

Based on isothermal titration calorimetry (ITC) measurements, ARM1, TTSe and TTO possess similar binding parameters (Figure 6A). The dissociation constants (K_d_) for ARM1, TTSe and TTO were determined as 0.25, 0.37 and 0.33 µM, respectively, which is in correlation with previous reports for ARM1 [29]. 

In addition, differential scanning fluorimetry (DSF) analysis was performed to assess the effects of selective inhibitors on the stability of LTA_4_H. The T_m_ values of LTA_4_H increased from 57.5 °C up to 61.2 °C, 61.9 °C and 60.7 °C upon the addition of 0 to 200 µM of ARM1, TTSe and TTO, respectively (Figure 6B). Elevated T_m_ values indicated that all inhibitors stabilize LTA_4_H. Based on the concentration–response curves (Figure 6B, right panel), the K_d_ values for ARM1, TTSe and TTO were determined as 0.24 ± 0.01 µM, 0.30 ± 0.02 µM and 0.51 ± 0.04 µM, respectively, which are in agreement with the K_d_ values observed with ITC.

### 2.6. Selective Inhibitors Stabilize LTA_4_H via Conformational Shift

Previously, LTA_4_H has been crystallized in a complex with different inhibitors, including ARM1, using a liquid/liquid diffusion in capillaries [20,30] or a sitting drop method [31]. In the current study, a more convenient hanging drop vapor diffusion technique was introduced for the co-crystallization. The current approach resulted in 1.42 Å and 1.35 Å structures of LTA_4_H with TTSe (PDB ID: 8AWH) and TTO (PDB ID: 8AVA), respectively (Appendix A).

Similarly to ARM1, TTSe and TTO bound to the end of the hydrophobic substrate channel and interact with the polypeptide chain of LTA_4_H via hydrogen–hydrogen interactions (Figure 7A). In addition, the hydrophilic moiety of TTSe and TTO was coordinated by two water molecules connected with the peptide bonds of A377 and S379 from one side or D312 and W315 from the other side (Figure 7B).

The hydrophilic selenazole or oxazole ring can rotate via a single C-C bond; therefore, densities of two rotamers are detected. TTSe and TTO-bound LTA_4_H structures are in a closed conformation which is in accordance with stabilizing conformational changes observed with DSF (Figure 7C). The binding of inhibitors results in the conformational rotations of Y383 and Y378 which together with catalytic Zn^2+^ coordinate the substrate for efficient catalysis (Figure 7D).

### 2.7. Predicted Bindings Sites of 5-LOX for Chalcogen-Containing Inhibitors

Docking simulations with chalcogen-containing inhibitors revealed that compounds can bind in between the C2-like domain and the catalytic domain (Binding Site 1) of the AF 5-LOX and in the catalytic center (Binding site 2) of the uncorked stable 5-LOX (PDB ID: 6N2W) with similar binding scores (Figure 8). These binding sites have been identified with AKBA (PDB ID: 6NCF) and NDGA (PDB ID: 6N2W), respectively [6]. Chalcogen-containing inhibitors were located at the Binding site 1 in two different conformations with the polar group facing either towards Gln130 (shown in Figure 8) or Arg102. At the Binding site 2, polar headgroups of inhibitors are faced towards the catalytic center with slightly different orientations between the thiazole, selenazole and oxazole moieties. This may also explain differences in the inhibition constants with recombinant 5-LOX.

## 3. Discussion

In the current study, the inhibitory and binding properties of different chalcogen-containing LTA_4_H inhibitors were characterized. The same chalcogen replacement strategy for inhibitors of other enzymes has been implemented in the past [26]. For instance, chalcogen alterations did not influence the potency of inhibitors on the enzymatic and cellular activity of tryptophan 2,3 dioxygenase but affected different physicochemical properties, such as lipophilicity, toxicity and microsomal stability. In the current study, predicted pharmacological properties for ARM1, TTSe and TTO indicated that they may have different bioavailability and biophysical–biochemical properties in vivo which is a matter of future research (Appendix A). Pure oxygen, sulfur and selenium are generally non-toxic; however, hydrogenated or oxidized derivatives of sulfur and selenium are harmful. Chalcogen-coupled nanoparticles indicated that sulfur is more toxic to nematodes compared to selenium [32]. In contrast, a methyl-thiazolyldiphenyl-tetrazolium (MTT) assay of chalcogenides in semiconductors revealed the higher toxicity of selenium [33]. Therefore, a toxicological assessment should be performed for new chalcogen-containing compounds. It is known that selenium can induce the production of reactive oxygen species in different biological systems [26,34]. Until today, more pharmacological compounds have been synthesized with the oxazole and thiazole rather than the selenazole moiety [35,36], including cyclooxygenase-2 inhibitors [37]. In general, thiazole compounds are more promising drug candidates than oxazole and selenazole derivatives. In the current study, ARM1, TTSe and TTO at lower concentrations (less than 30 µM) were tested as safe compounds in human leukocytes and there were no apparent chalcogen-specific effects observed.

TTSe, TTO and ARM1 bind to the substrate channel of LTA_4_H and induce conformational changes which stabilize LTA_4_H (Figure 6B and Figure 7C). This phenomenon was also observed with an inactive mutant, LTA_4_H D375N, that was crystallized either in an open conformation or a closed conformation with the LTA_4_ substrate [19]. However, the importance of conformational and regulatory aspects of LTA_4_H in intact cells remains to be studied. In vitro (Figure 4) and in silico (Figure 5) results suggest that the closed conformation in the presence of a ligand may increase the aminopeptidase activity of LTA_4_H. Selective inhibitors or other ligands can be further developed to induce increased degradation of naturally occurring PGP. Furthermore, TTSe, TTO and ARM1 are bound to the substrate channel of LTA_4_H in two different conformers. This creates an opportunity to design more rigid ARM1 derivatives with selective hydrogen bonding which could improve the potency of these compounds. It is supported by molecular dynamics simulations which indicate that currently available selective LTA_4_H inhibitors may be too mobile and possess multiple binding poses [23]. The inhibition of 5-LOX by chalcogen-containing inhibitors is supported by the docking simulations, indicating the binding of these compounds to regions relevant to enzyme inhibition. However, the structural characterization of inhibitor binding to 5-LOX requires further investigation.

Even though ARM1, TTSe and TTO contain different chalcogens with distinct atomic radii, redox potential and electronegativity (Appendix A), the inhibitory and binding properties of these selective inhibitors with LTA_4_H were rather similar. In contrast, results from the activity assay with recombinant 5-LOX showed distinct chalcogen-dependent inhibition which was not that evident with intact cells. This indicates that recombinant 5-LOX is more sensitive to chalcogen replacements than intracellular 5-LOX which can be explained with the dual inhibition of LTA_4_H and 5-LOX in cells. Similar inhibition parameters with the three compounds with LTA_4_H may suggest that the hydrophobic interactions between the benzylphenyl moiety of an inhibitor and hydrophobic residues in LTA_4_H decide the binding efficiency. This is consistent with the structures of LTA_4_H co-crystallized with other benzylphenyl-containing inhibitors, 4-MDM [38] and 4-OMe-ARM1 [24]. Although 4-OMe-ARM1 shares a similar binding mode to ARM1, TTSe and TTO, the 4-MDM compound has the hydrogen–hydrogen interaction only between O of the methoxy group and Q136 in the middle of the substrate channel (Appendix A). It can be postulated that 4-MDM has the lowest intracellular selectivity due to the smaller size and higher hydrophobicity. Until today, there have no comparative studies been performed to assess the efficacy of 4-MDM, ARM1-type inhibitors and resveratrol-type compounds in different disease models. It has been shown that LTA_4_H helps to prevent the accumulation of pro-inflammatory PGP in lungs [14,15,38]. Even though LTA_4_H-deficient mice lack both activities, it is not clear if higher levels of PGP in the absence of LTA_4_H are due to the lack of peptidase activity of LTA_4_H or compensating effects of the immune system [23]. Therefore, more studies are needed to understand the biological importance of the aminopeptidase activity of LTA_4_H at the molecular level. Today, only LYS006 [39] and Acebilustat [40] as general LTA_4_H inhibitors have passed to phase II clinical trials. However, we lack information about drug candidates that spare the peptidase activity of LTA_4_H.

Simultaneous inhibition of 5-LOX and LTA_4_H can be a new strategy to modulate the production of pro-inflammatory mediators. As the inhibition of 5-LOX by TTSe and TTO is weaker, these inhibitors can be used to selectively block the formation of LTB_4_, whereas ARM1 can target the formation of 5*S*-HETE as well as LTB_4_. Unaffected or weaker inhibition of 5-LOX may retain the capacity to produce pro-resolving lipoxins by 5-LOX together with 15-LOX.

## 4. Materials and Methods

### 4.1. Materials 

Inhibitors, 4-(4-benzylphenyl) thiazol-2-amine (ARM1), 4-(4-benzylphenyl) selenazol-2-amine (TTSe) and 4-(4-benzylphenyl) oxazol-2-amine (TTO) were synthesized by LifeChemical Co (Appendix A). A23187, AA, prostaglandin B_1_ and B_2_ (PGB_1_ and PGB_2_), phenyl-cyano-(6-methoxy-2-naphthalenyl)methyl ester-2-oxiraneacetic acid (PHOME), different fatty acid standards and the PGP peptide was purchased from Cayman Chemicals. High-performance liquid chromatography (HPLC)-grade methanol and trifluoracetic acid were from VWR (Darmstadt, Germany). Cytotox96^®^ non-radioactive cytotoxicity assay was purchased from Promega™ Corporation (Madison, WI, USA). Dulbecco’s Buffer Substance (PBS), SERVA Electrophoresis (Heidelberg, Germany), ATP-agarose, dextrane, fetal calf serum (FCS), Histopaque^®^-1077, RPMI 1640, staurosporine, 3-(4,5-dimethylthiazol-2-yl)-2,5-diphenyltetrazolium bromide (MTT) were from Merck KGaA (Darmstadt, Germany). The methyl ester of LTA_4_ was purchased from Med Chem 101 and saponified in tetrahydrofuran with 1 M LiOH [6% (*v/v*)] for 48 h at 4 °C. Unless mentioned otherwise, solvents and all other reagents were obtained from Carl Roth (Karlsruhe, Germany) or Sigma-Aldrich (Schnelldorf, Germany).

### 4.2. Expression and Preparation of Human Recombinant 5-LOX and LTA_4_H

*Escherichia coli* BL21 cells were transformed with the pT3-5-LO vector and human recombinant 5-LOX was expressed overnight at 30 °C as previously described [41]. Cells were lysed in 50 mM Tris-HCl pH 7.5, 200 mM NaCl, 5% (*v*/*v*) glycerol, 1 mM EDTA, 1 mM phenylmethanesulphonyl fluoride, 60 µg/mL soybean trypsin inhibitor, 1 mg/mL lysozyme and homogenized by sonication (3 × 20 s) followed by centrifugation at 13,000× *g* for 45 min at 4 °C. The supernatant was loaded on an ATP-agarose column and washed with 50 mM phosphate buffer (PB) containing 1 mM EDTA. Protein fractions were eluted with PB supplemented with 1 mM EDTA and 20 mM ATP. Aliquots of semi-purified 5-LOX were diluted with PBS containing 1 mM EDTA and used immediately in incubations.

Human recombinant LTA_4_H was expressed in *Escherichia coli* JM101 strain and purified as described previously [42]. In short, supernatant of lysed cells was loaded on a nickel affinity column followed by the purification steps with the Mono Q 5/50 and Superdex HiLoad 16/600 columns (Cytiva, Uppsala, Sweden). Protein purity was checked by sodium dodecyl-sulfate polyacrylamide gel electrophoresis (SDS-PAGE)(XCell SureLock Mini-Cell Electrophoresis System and PowerEase 300W Power Supply) with a NuPAGE Bis-Tris 4–12% gels (Invitrogen, Waltham, MA, USA) after every purification step. Protein concentration was determined by UV absorbance at 280 nm (Varian Cary 300 Bio UV-VIS spectrometer) using an extinction coefficient of 104,905 M^−1^ cm^−1^ for human LTA_4_H (Appendix A).

The human recombinant soluble epoxide hydrolase (sEH) was expressed and purified as described before [43].

### 4.3. Inhibition Assays with Recombinant Enzymes

Aliquots of 5-LOX (0.5 μg) were pre-incubated with test compounds or a vehicle (0.1%, *v*/*v*) for 10 min at 4 °C. Then, samples were stimulated with 2 mM CaCl_2_ and 20 µM AA to induce the 5-LOX activity at 37 °C. The reaction was stopped after 10 min by addition of 1 vol of ice-cold methanol. Formed metabolites, i.e., LTB_4_, trans- and epi-trans-LTB_4_, and 5*S*-HETE, were analyzed with reverse-phase HPLC using a C18 RP Radial PAK column (Waters) as described previously [44].

Purified LTA_4_H (3 μg) was pre-incubated with 0–50 μM inhibitors on ice for 3 min and the epoxide hydrolase reaction was carried out with 10 μM, 20 μM or 40 μM LTA_4_ for 15 s in 100 μL 25 mM Tris-HCl pH 7.8 at room temperature. The incubation was stopped by addition of 2 vol of methanol, containing 300 pmol PGB_2_ (ε ~ 26,000 M^−1^ cm^−1^; λ_max_ = 280 nm) as an internal standard, and followed by 1 vol of distilled water. Samples were analyzed by reverse-phase HPLC on a 3.9 × 150-mm C18 Nova-Pak column (Waters) eluted with acetonitrile/methanol/water/acetic acid at a ratio of 30:36:34:0.1 (*v/v*) and at a flow rate of 1 mL/min. The formation of LTB_4_ (ε ~ 50,000 M^−1^ cm^−1^) was determined at 270 nm. The K_i_ value for each inhibitor was calculated using the competitive inhibition model in the GraphPad Prism 9 program.

Pre-incubation with 0.5 µg/mL of sEH and test compounds or 0.1% (*v*/*v*) vehicle was performed in 25 mM Tris-HCl buffer (pH 7.0) containing 0.1 mg/mL bovine serum albumin at room temperature for 10 min. Reaction was initiated with 50 μM PHOME, a non-fluorescent compound that is enzymatically converted into fluorescent 6-methoxy-naphtaldehyde, at room temperature for 60 min. Reactions were stopped with 200 mM ZnSO_4_ and fluorescence signals were recorded at 330 nm/465 nm (ex/em). If required, a possible fluorescence of test compounds was subtracted from the read-out.

### 4.4. Peptidase Assay with LTA_4_H

The effect of inhibitors on the aminopeptidase activity of LTA_4_H was determined with 0.5 μg LTA_4_H, 0–100 μM of each inhibitor and 800 μM PGP in 100 μL 10 mM Tris-HCl pH 7.8 for 3 min at 37 °C. The reaction was stopped by addition of 150 μL of acetic acid. To detect and quantify released N-terminal proline, incubations with 150 μL of ninhydrin solution (25 mg/mL) were carried out at 100 °C for 45 min. Samples were cooled down and purple-colored ninhydrin conjugates were extracted with 350 μL toluene and the upper phase was transferred to a polypropylene 96-well plate (Greiner Bio-One, Kremsmünster, Austria). All incubations were carried out in triplicates and absorbance was measured at 495 nm using a TECAN Infinite M200 plate reader. Specific peptidase activities were determined based on the standard curve of free L-proline (Appendix A). 

In addition, the aminopeptidase assay was performed with p-nitroanilide (pNa) and 7-amido-4-methylcoumarin (AMC) conjugates containing different N-terminal amino acids. Incubations were carried out with 1 μg LTA_4_H and 1 mM pNa substrates or 5 µM AMC substrates in 400 μL 10 mM Tris-HCl pH 7.8 containing 100 mM KCl at room temperature for 30 min. The absorbance of released pNa was measured after every 5 min at 405 nm using a TECAN Infinite M200 plate reader. The measurements with AMC derivatives were carried out using a black polystyrene 96-well plate and the fluorescence of released AMC at 360 nm/460 nm (ex/em) was recorded. Specific activities were calculated based on the standard curves of free pNA and AMC (Appendix A). The activation (AC_50_) or inhibition constants (IC_50_) were determined using the non-linear regression approach in the GraphPad Prism 9 program.

### 4.5. Preparation of Human Primary Leukocytes

Human primary leukocytes were isolated from peripheral blood of healthy adult volunteers provided by the Institute of Transfusion Medicine at the University Hospital Greifswald as described before [45]. All methods were performed in accordance with the Declaration of Helsinki. In brief, erythrocytes were removed by dextran sedimentation and leukocytes were separated by the density gradient centrifugation on a lymphocyte separation medium (Histopaque^®^-1077, Merck, Darmstadt, Germany). The remaining erythrocytes were removed by hypotonic lysis using water, and resulting polymorphonuclear neutrophils (PMNs) were resuspended in PBS containing 0.1% (*w*/*v*) glucose (PG buffer) or PG buffer with 1 mM CaCl_2_ (PGC buffer) as indicated. Resulting peripheral blood mononuclear cells (PBMC), including monocytes, were seeded in the RPMI 1640 medium supplemented with 10% (*v*/*v*) FCS, 100 U/mL penicillin, 100 µg/mL streptomycin and 2 mM L-glutamine in cell culture flasks (Greiner Bio-one, Frickenhausen, Germany) for 1.5 h at 37 °C and 5% CO_2_. Adherent monocytes were washed twice with PBS and were finally resuspended in the RPMI 1640 medium as described before [45].

### 4.6. Cytotoxicity Assays and Prediction of Physico-Chemical Properties of Inhibitors

Monocytes (200,000 cells/well in 100 µL of RPMI 1640 containing 10% FCS, 100 U/mL penicillin, 100 µg/mL streptomycin and 2 mM L-glutamine) were seeded in a 96-well plate. Cells were allowed to adhere for 1.5 h at 37 °C and 5% CO_2_. Cells were incubated at 37 °C and 5% CO_2_ with the vehicle (0.5%, *v*/*v*) or compounds for 24 h. Then, cells were incubated with 5 mg/mL MTT in PBS until the blue staining of the vehicle-containing control cells. Formazan formation was stopped by 100 µL of 10% (*w*/*v* in 20 mM HCl) SDS lysis buffer and followed by shaking overnight. Finally, absorbance was measured at 570 nm with a SpectraMax^®^ i3x multi-mode detection platform (Molecular Devices, San Jose, CA, USA). The pan-protein kinase inhibitor staurosporine (1 µM) was used as a cytotoxic control inhibitor.

For analysis of extracellular lactate dehydrogenase (LDH) as a plasma membrane integrity marker, the CytoTox96^®^ non-radioactive cytotoxicity assay kit was used. Freshly prepared PMNs (10^6^/mL in PG buffer) were seeded in a 96-well plate. Cells were treated with the vehicle (0.5%, *v*/*v*), Triton X-100 (10%, *v*/*v*) for the lysis control or compounds for 30 min. After the incubation, the LDH release was measured by recording the absorbance at 490 nm with a SpectraMax^®^ i3x multi-mode detection platform (Molecular Devices, San Jose, CA, USA).

In addition, physicochemical properties of ARM1, TTSe and TTO were predicted using the Osiris Property Explorer and SwissADME tools [46]. The logD values at different pH were calculated using the LogD Predictor tool (ChemAxon, Budapest, Hungary). 

### 4.7. 5-LOX Product Formation from Intact PMNs

PMNs (5 × 10^6^ cells/mL) were diluted in PGC buffer and pre-incubated with the vehicle (0.1%; *v*/*v*) or compounds for 10 min at 37 °C prior to the stimulation with 2.5 µM A23187. After 10 min at 37 °C, treatments were stopped on ice with an equal volume of ice-cold methanol. Next, 530 µL acidified PBS containing 200 ng PGB_1_ as an internal standard were added to the sample followed by the solid phase extraction with on a RP18 column. In a similar manner, treatments with intact PMNs, 0–20 µM AA and 0.5 µM or 10 µM ARM1, TTSe or TTO were performed to see the effect of exogenous AA on the inhibition by chalcogen-containing compounds. Formed 5-LOX metabolites were analyzed as described previously [47].

### 4.8. Determination of Binding Affinity with Isothermal Titration Calorimetry

ITC was performed with an iTC200 calorimeter (MicroCal Inc., Northampton, MA, USA) equipped with a 0.3 mL sample cell. Measurements were performed in 25 mM Tris-HCl pH 7.8 supplemented with 3% DMSO at 25 °C. Due to the limited solubility of selective inhibitors, 15 µM ARM1, 15 µM TTSe or 20 µM TTO was placed in the sample cell instead of the syringe and titrated with 100 µM LTA_4_H [29]. After the ITC system was equilibrated, the initial injection of an inhibitor was performed in 0.5–1.0 µL followed by 15 injections in 2.5 µL. The time between each injection was 1.5–3.0 min to ensure the complete equilibration of the system. Collected data were analyzed by MicroCal PEAK-ITC Analysis Software. Control experiments were performed with identical conditions using only inhibitor, protein or buffer with 3% DMSO.

### 4.9. Thermal Shift Assay with Differential Scanning Fluorimetry

DSF, also known as the thermal shift assay, was carried out in 25 mM Tris-HCl pH 7.8 containing 50 ng/µL LTA_4_H, 2X SYPRO Orange (Life Technologies, Carlsbad, CA, USA), 3% DMSO or 0–300 µM of an inhibitor. The assay was performed in a white 96-well Multiplate PCR Plate (Bio-Rad, Hercules, CA, USA) covered with the iCycler iQ Optical tape (Bio-Rad, Hercules, CA, USA). The total volume per well was 25 μL, consisting of 1 μL SYPRO Orange solution. Relative fluorescence intensities (RFU) were recorded with a C1000 Touch Thermal Cycler (Bio-Rad, Hercules, CA, USA) and a CFX-96 Real-Time System. The temperature was raised from 20 to 90 °C by 0.2 °C in 15.8 s per step. Raw data from melting curves were exported from the CFX manager software to the Thermott data analysis platform (www.thermott.com; accessed on 18 April 2023) [48] and presented using the GraphPad Prism program. The melting temperature (T_m_) for each condition were obtained from the first derivative of melting curves and the affinity (K_d_) and stability (K_b_) values were calculated based on the T_m_ and concentration–response curves.

### 4.10. Co-Crystallization of LTA_4_H with Inhibitors 

Human recombinant LTA_4_H (10 mg/mL) in 25 mM Tris-HCl pH 7.8 was co-crystallized with 0.5 mM TTSe or TTO using the hanging drop vapor diffusion approach. In brief, 1 μL protein solution with 0.5 mM of an inhibitor was incubated with 1 μL precipitant solution composed of 12% (*w*/*v*) PEG 8000, 100 mM NaAc, 100 mM imidazole buffer pH 6.8, 5 mM YbCl_3_ and 0.5 mM of each inhibitor on a siliconized coverslip using the Greiner 24-well pre-greased plates at 21 °C. The plate-shaped crystals in clusters appeared within 7 days. Stacked crystals were crushed and microcrystals were seeded using the same co-crystallization approach. Single plate-shaped crystals appeared within 7 days at 21 °C. Next, individual crystals were soaked with 0.5 mM of each inhibitor for 2 h and collected with the CrystalCap ALS HT cryoloops (Hampton Research, Aliso Viejo, CA, USA) in the cryoprotecting solution consisting of the crystallizing solution and 30% (*v*/*v*) glycerol prior to immediate freezing in liquid nitrogen.

### 4.11. Data Collection and Processing

Crystallization data was collected under cryogenic conditions (100 K) at a wavelength of 12.7 KeV (0. 9763 Å) at the BioMAX beamline in MAX IV Laboratory (Lund, Sweden). For each diffraction dataset, 3600 images at a resolution of 1.5 Å with an oscillation angle of 1° per image and a crystal-to-detector distance of 197 mm were collected. Data for crystals were indexed, integrated, and scaled using XDSAPP [49]. The resolution of datasets was corrected based on the cc(1/2) coefficient and completeness criteria. Collection parameters and statistics for each dataset are summarized in Appendix A.

### 4.12. Structure Solution, Refinement and Validation

Datasets with TTSe and TTO were prepared using the Staraniso server (https://staraniso.globalphasing.org/; 18.04.2024) using anisotropic approach. Datasets were cleaned and prepared with MRFANA by the Staraniso approach and with *Uniqueify* (or import) in the CCP4i system. For both datasets, the molecular replacement was carried out with PHASER [50] using the coordinates of human LTA_4_H in a complex with ARM1 (PDB ID code 4L2L) [20] as the search model. All structures were initially refined with REFMAC5 in CCP4i [51] and submitted to the TLS server (http://skuld.bmsc.washington.edu/~tlsmd/; accessed on 18 April 2023). The output files from TLS were used in the final refinement with REFMAC5. The PDB and Crystallographic Information File formats were generated for all ligands using the AceDRG tool [52] in CCP4i. Atomic displacement parameters were modelled using a combination of anisotropic/isotropic approaches. Each model was inspected manually and validated using MolProbity [53] and other Coot [54] validation functions. Final models were deposited in the PDB databank (www.rcsb.org; 18.04.2023). All figures presenting protein structures were made using the Chimera 1.15 software (San Francisco, CA, USA) [55].

### 4.13. Molecular Docking Simulations

Potential interaction sites between 5-LOX and inhibitors were predicted via docking simulations using the AlphaFold-generated 5-LOX and the uncorked 5-LOX (PDB ID: 6N2W) as templates. To understand the effect of ARM1 on the peptidolysis of different peptide analogues, peptide analogues as ligands were docked into the crystal structure of LTA_4_H in a complex with ARM1 and a stable PGP derivative, 1-[4-oxo-4-[(2S)-pyrrolidin-2-yl]butanoyl]-L-proline (OPB-Pro) (PDB ID: 4MKT), LTA_4_H D375N in open conformation with open (PDB ID: 5NID) and closed (PDB ID: 5NIA) tyrosine gates. Peptide analogues were prepared using the Grade Web Server (https://www.globalphasing.com; accessed on 18 April 2023). Prior to the docking, all ligands, except catalytic metals, were removed from the X-ray structures. Simulations were carried out with the AutoDock Vina tool [56]. Binding modes with the best docking scores were visualized using the Chimera software.

### 4.14. Statistical Analysis

Data were presented as mean ± SEM unless otherwise specified. Differences among groups were evaluated by Student’s *t*-test, One-way or Two-way ANOVA and a value of *p* < 0.05 was considered statistically significant.

## 5. Conclusions

In summary, chalcogen-containing inhibitors are potential lead molecules that can be improved to selectively spare and possibly increase the aminopeptidase activity of LTA_4_H. These inhibitors possess a potential to modulate the immune system and inflammatory processes through inhibition of the 5-LOX and LTA_4_H enzymatic cascade.

## Figures and Tables

**Figure 1 ijms-24-07539-f001:**
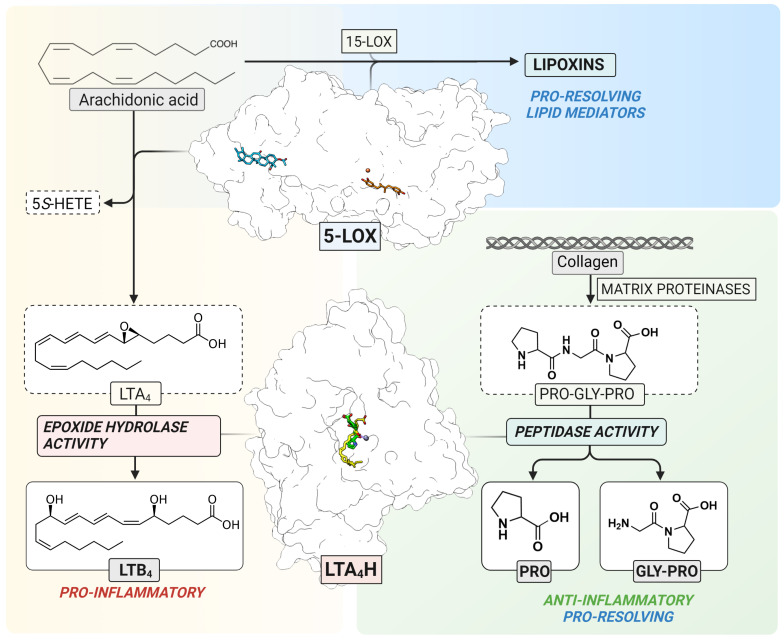
The 5-LOX pathway. 5-LOX catalyzes the oxygenation of arachidonic acid to 5*S*-HETE and the LTA_4_ epoxide as the substrate for LTA_4_H. In parallel, 5-LOX contributes to the formation of lipoxins in cooperation with 15-LOX (**top**). LTA_4_H catalyzes the conversion of unstable LTA_4_ to LTB_4_ (**left**) or the cleavage of a neutrophil chemoattractant, Pro-Gly-Pro (**right**). The structure of 5-LOX has been elucidated with two inhibitors, NDGA (orange) in the catalytic center containing non-heme iron (red) and AKBA (blue) inside an allosteric pocket between the β-barrel and catalytic domain, respectively. The structure of LTA_4_H contains overlapping binding sites for LTA_4_ (yellow) and Pro-Gly-Pro (green) which interact with the catalytic zinc (purple).

**Figure 2 ijms-24-07539-f002:**
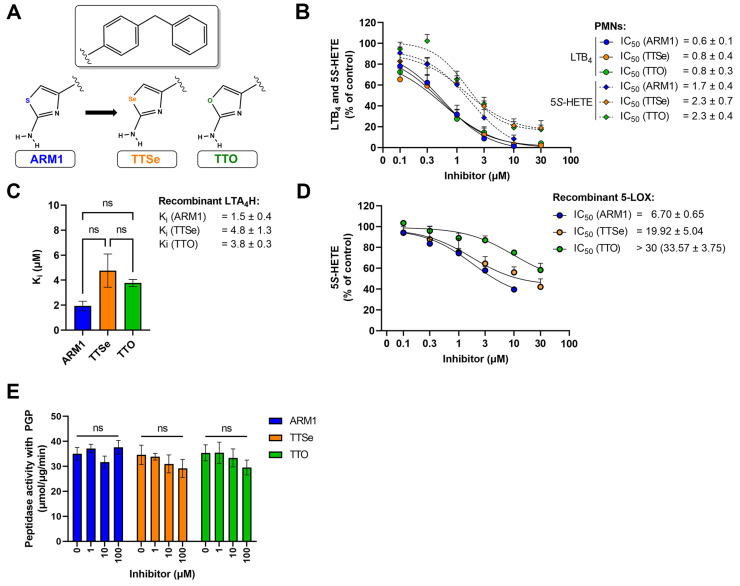
Inhibition assays with chalcogen-containing inhibitors. (**A**) The chemical structures of inhibitors: ARM1, TTSe and TTO; (**B**) The IC_50_ values of chalcogen-containing inhibitors determined based on the cellular activity of LTA_4_H and 5−LOX in activated polymorphonuclear neutrophils (PMNs) (**C**) The inhibition constants (K_i_) of inhibitors determined with recombinant human LTA_4_H; (**D**) The IC_50_ values of inhibitors with recombinant human 5−LOX; (**E**) The effect of inhibitors on the aminopeptidase activity of LTA_4_H with the PGP substrate. ns—non−significantly different values.

**Figure 3 ijms-24-07539-f003:**
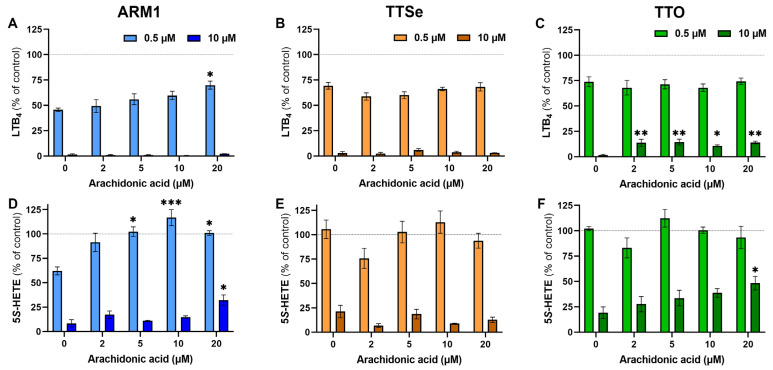
The effect of arachidonic acid at different concentrations on the potency of inhibitors. (**A**–**C**) The levels of LTB_4_ were determined from intact PMNs treated with 0–20 µM of arachidonic acid in the presence of 0.5 µM or 10 µM of ARM1, TTSe or TTO. (**D**–**F**) The levels of 5*S*-HETE were determined in parallel. Statistically significant values were derived in comparison with the control value in the absence of exogenous AA. * *p* value < 0.05; ** *p* value < 0.01; *** *p* value < 0.001; non-significant differences are not indicated.

**Figure 4 ijms-24-07539-f004:**
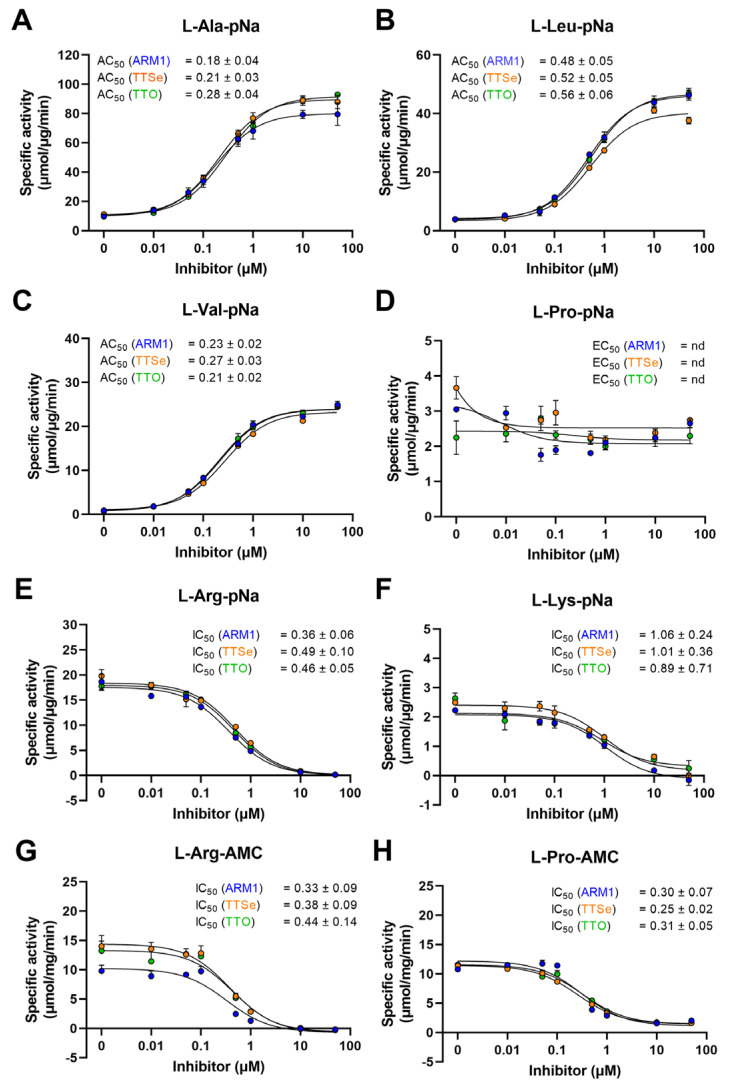
Effect of chalcogen-containing inhibitors on the peptidase activity of LTA_4_H. (**A**–**C**) Specific activities of LTA_4_H determined with hydrophobic pNa conjugates; (**D**) The specific activity with L-Proline-pNa; (**E**,**F**) Activity measurements with substrates containing positively charged longer residues; (**G**,**H**) The IC_50_ values determined with AMC derivatives.

**Figure 5 ijms-24-07539-f005:**
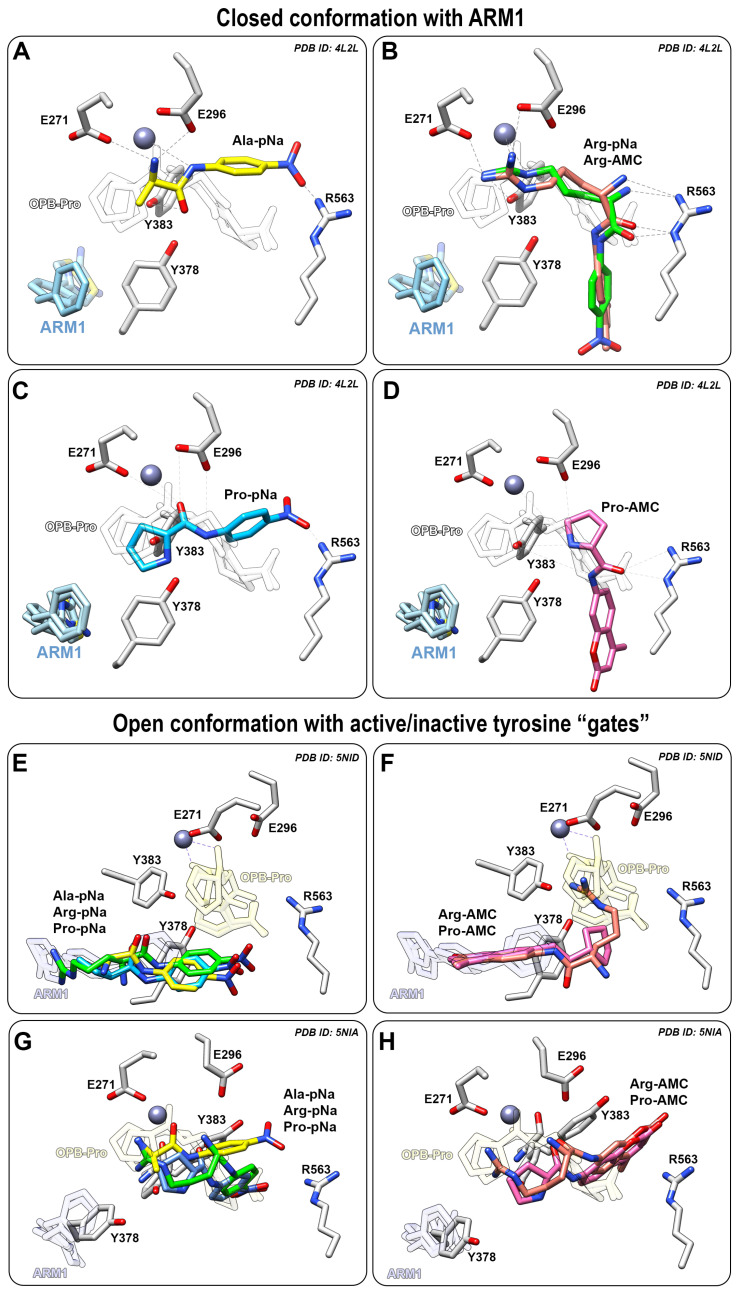
Peptide analogues docked into LTA_4_H or the ARM1-bound LTA_4_H. (**A**−**D**) Binding modes of Ala−pNa (yellow, **A**), Arg−pNa (green) or Arg−AMC (pink, **B**), Pro−pNa (blue, **C**) or Pro−AMC (pink, **D**) in LTA_4_H bound with ARM1 (PDB ID: 4L2L); (**E**,**F**) The binding mode of Ala−pNa, Arg−pNa, Pro−pNa (**E**) and Arg−AMC, Pro−AMC (**F**) in LTA_4_H D375N in an open conformation with inactive (**E**,**F**) or active (**G**,**H**) tyrosine “gate”.

**Figure 6 ijms-24-07539-f006:**
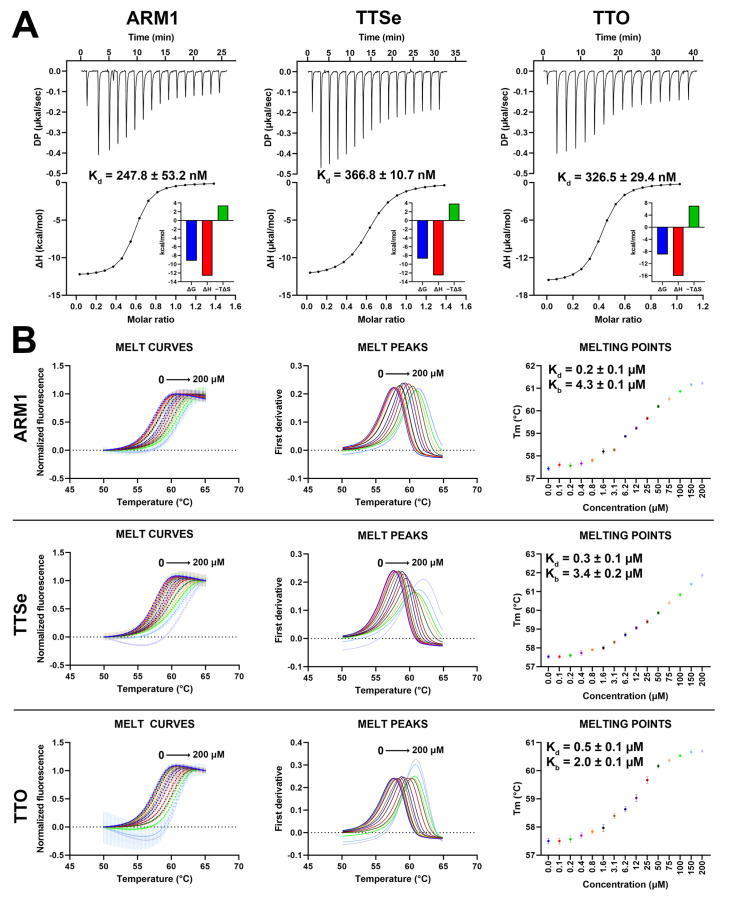
Binding parameters of chalcogen-containing inhibitors with LTA_4_H. (**A**) Isothermal calorimetric analysis (ITC) of 100 µM LTA_4_H with 15 µM ARM1, 15 µM TTSe and 20 µM TTO in 25 mM Tris pH 7.8 with 3% DMSO at 25 °C. The dissociation constants (K_d_) were determined based on the titration curves. Data are presented as mean ± SEM with n = 4. (**B**) The thermal stability of LTA_4_H by selective inhibitors assessed by differential scanning fluorimetry. Different colors represent different inhibitor concentrations. The dissociation (K_d_) and stability (K_b_) constants were determined based on the dose−response curves.

**Figure 7 ijms-24-07539-f007:**
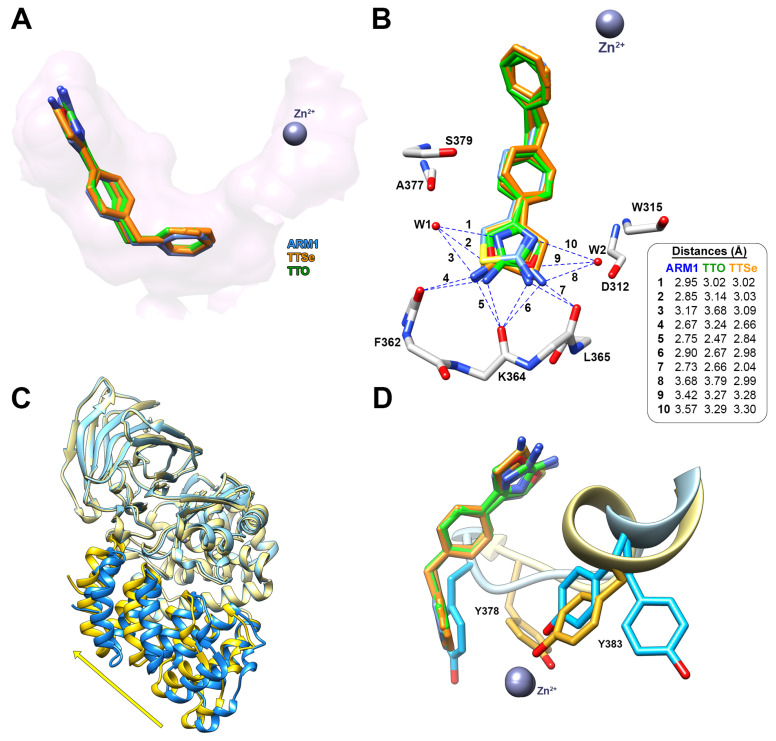
The binding mode of TTSe and TTO in LTA_4_H. (**A**) TTSe (orange), TTO (green) and ARM1 (blue) are bound to the end of the substrate channel of LTA_4_H; (**B**) Distances of hydrogen bonds between inhibitors, the polypeptide chain of LTA_4_H and water molecules (W1 and W2); (**C**) The domain movement from open (blue; PDB ID: 5NIA) to closed conformation (yellow) induced by selective inhibitors; (**D**) the inhibitor-induced rotation of Y378 and Y388 into the activated state of LTA_4_H.

**Figure 8 ijms-24-07539-f008:**
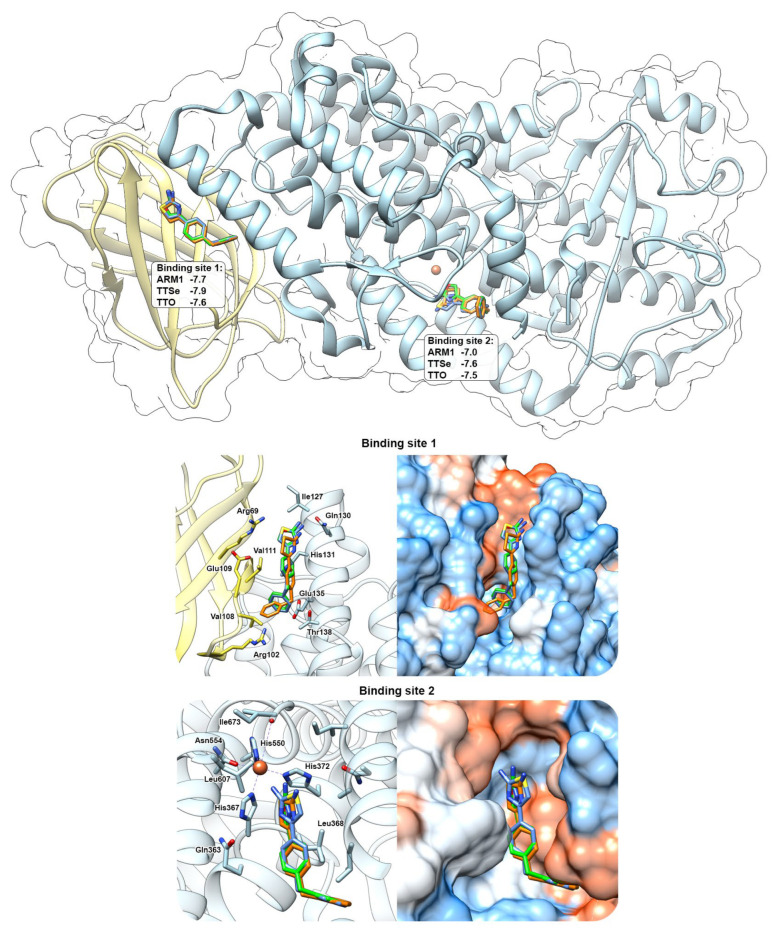
In silico prediction of the interaction between chalcogen−containing inhibitors and 5−LOX. ARM1 (blue), TTSe (orange) and TTO (green) were docked to the pocket between the C2−like domain (yellow) and the catalytic domain (blue) (Binding site 1) and inside the catalytic center (Binding site 2). Docking simulations were carried out with the AlphaFold−generated 5−LOX and uncorked 5−LOX (PDB ID: 6N2W) and the binding scores for each inhibitor were determined (top panel).

## Data Availability

The data presented in this study are available within the article and its Appendix A. In addition, the protein structures are deposited in the PDB database.

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
