# Peer review of "Modulation of the 5-Lipoxygenase Pathway by Chalcogen-Containing Inhibitors of Leukotriene A4 Hydrolase"

_ijms, 2023, doi:10.3390/ijms24087539_

Round 1

Reviewer 1 Report

The authors present a very nice study and suggest that "the inhibitors possess a potential to modulate the immune system and inflammatory processes through inhibition of the 5-LOX and LTA4H enzymatic cascade", which represents a promising intervention for disease associated with immunopathology. However the interplay of pro-inflammatory and anti-inflammatory mediators is very complex and it will be interesting to test these compound in a model of e.g. infectious or autoimmune disease.

My comments:

- the abbreviation PMN is not introduced

- is the inhibitor species specific?

- although the data are very convincing, it would be nice to see whether the inhibitors have an effect on PMN LTA4H knockout cells (which are available for the murine system).

- is there toxicity on other cells (epithelial cells e.g.)

- functional testing - how is the inflammatory response of cells generally modified by the inhibitors?

Author Response

The authors are grateful for the kind feedback on our work. 

1. The abbreviation PMN is not introduced.

The abbreviation has now been introduced and used throughout the text.

2.  Is the inhibitor species specific?

These inhibitors have been tested only on human cells or human recombinant enzymes. Therefore, we lack information about species-specificity. However, as 5-LOX and LTA4H are highly conserved in mammals, we would expect similar inhibition across all pharmacologically relevant animal models.

3. Although the data are very convincing, it would be nice to see whether the inhibitors have an effect on PMN LTA4H knockout cells (which are available for the murine system).

We agree with this reviewer that testing of these inhibitors in LTA4H-deficient PMN would be interesting. However, at present we do not have LTA4H knock-out mice and to establish these animals at Karolinska Institutet is a very time consuming and cumbersome process. Hence, it will not be possible to conduct such experiments in a timely manner. 

4. Is there toxicity on other cells (epithelial cells e.g.)?

Based on the cytotoxicity profiles on leukocytes, we expect a similar outcome with other cell types. Again, the time limit for resubmission (10 days) does not permit additional experiments with new cell lines, which requires establishment of new cultures. 

5. Functional testing - how is the inflammatory response of cells generally modified by the inhibitors?

The role of 5-LOX in the inflammatory response and immune modulating properties of LTB4 are very well established in isolated cells and various preclinical models of inflammation (reviewed by Wan et al. 2017). LTB4 is an extremely potent pro-inflammatory chemotactic compound of neutrophils and reduced levels of LTB4 result in lower migration of surrounding cells into inflammatory tissue. In addition, LTB4 promotes phagocytosis of different myeloid cells that is also diminished by lower levels of LTB4. From this vast literature, we infer that our inhibitors have the potential to reduce inflammation in LTB4 dependent models. To address this point, sections of the text have revised to reflect the fact that we have not included animal models of inflammation in our work. Please see page 1, lines 27-28; page 3, lines 106-108.

Reviewer 2 Report

The experimental design, results and discussion are appropriate and adequately described

Author Response

The authors are most grateful for the positive remarks by reviewer 2.

Reviewer 3 Report

Teder et al., present the manuscript, which title is Modulation of the 5-lipoxgenase pathway by chalcogen-containing inhibitors of leukotriene A4 hydrolase, is interesting. However, there are several questions in the manuscript, as below.

1. The authors should be improved the title of figure 1. And the authors should reorganization the relationship between the introduction and figure 1.

2. The authors should rewrite the section of materials due to the are some typo and format errors.

3. The authors should provide the number of IRB (Institutional Review Board).

4. The authors should provide the evidence and reason about the SAR (structure active relationship) on the 4-(4-benzylphenyl) selenazol-2-amine (TTSe) and 4-(4-benzylphenyl) oxazole-2-amine (TTO) were synthesized based on the lead compound, ARM1 in the present study.

5. Based on present results could not support their hypothesis including modulation of inflammation and inhibitor of LTB4. In figure 4, generation of LTB4 are differential results in the AA treatment at various concentration.

Author Response

The authors are grateful for the feedback on our work.

1. The authors should be improved the title of figure 1. And the authors should reorganization the relationship between the introduction and figure 1.

Thank you for pointing to this weakness of Figure 1 and introduction. The title of Figure 1 is now changed and hopefully improved. The introduction text has been scrutinized to assure coherence with figure 1. Please see page 2, line 52; pages 1-2, line 45-47.

2. The authors should rewrite the section of materials due to the are some typo and format errors.

The materials and methods section has been revised and corrected.

3. The authors should provide the number of IRB (Institutional Review Board).

Karolinska Institutet does not have formalized Institutional Review Boards. Nevertheless, internal revision has been conducted by all co-authors and researchers presented in the Acknowledgement.

4. The authors should provide the evidence and reason about the SAR (structure active relationship) on the 4-(4-benzylphenyl) selenazol-2-amine (TTSe) and 4-(4-benzylphenyl) oxazole-2-amine (TTO) were synthesized based on the lead compound, ARM1 in the present study.

Clarification of the rationale for the synthesis of chalcogen-containing inhibitors has been added to the Introduction (page 3, lines 100-106). Overall, ARM1 was developed based on the in silico predictions whereas new inhibitors, TTSe and TTO, were not designed using the SAR approach. Instead, the general idea was to perform chalcogen replacement, which may alter the hydrogen bonding network, and test the inhibitory and binding properties of new inhibitors. Effects of chalcogens and other chalcogen-containing compounds, including drug candidates, were also discussed in the Discussion section (pages 14-15, lines 334-351).

5. Based on present results could not support their hypothesis including modulation of inflammation and inhibitor of LTB4. In figure 4, generation of LTB4 are differential results in the AA treatment at various concentration.

The role of 5-LOX in the inflammatory response and immune modulating properties of LTB4 are very well established (reviewed by Wan et al. 2017). Nevertheless, the text has been changed to avoid overstatements of conclusions regarding modulation of inflammation by chalcogen-containing inhibitors. Please see page 1, lines 27-28; page 20, lines 611-612.

Presumably, this reviewer means Figure 3 instead of Figure 4 because AA treatment at various concentrations is presented in Figure 3. The graphs represent the ratios (in percentages) between untreated and inhibitor-treated intact PMNs. For instance, 100% or 0% means that the activity is either unaltered or completely blocked by an inhibitor. Absolute values for Figure 3 are presented in the Supplementary Figure S5.

Panels A-C show that at the lower concentration of inhibitors, 0.5 uM, the production of LTB4 was blocked around 25-50% whereas at the higher concentration, 10 uM, almost complete inhibition of LTA4H was achieved. Hence, these inhibitors work well already at submicromolar concentrations, even in the presence of high concentrations of AA.

Panels D-F show that 0.5 uM of inhibitors may not be sufficient to inhibit 5-LOX activity in intact PMNs but at the higher concentration, 10 uM, production of 5S-HETE was blocked at least 50%.

Even though there are slight fluctuations between treatments, probably due to unstable behavior of PMNs, the effect of these inhibitors is quite clear.

Overall, Figure 3 indicates that chalcogen-containing inhibitors block LTA4H more strongly, while higher concentrations of these compounds are needed to inhibit 5-LOX. Differences between the inhibition of 5-LOX and LTA4H observed in Figure 3 is also in agreement with Figure 2B-2D.

Round 2

Reviewer 3 Report

No more question.